# Optimization of Osmotic Dehydration of White Mushrooms by Response Surface Methodology for Shelf-Life Extension and Quality Improvement of Frozen End-Products

**DOI:** 10.3390/foods11152354

**Published:** 2022-08-06

**Authors:** Natalia A. Stavropoulou, Vassilis-Aggelos Pavlidis, Maria C. Giannakourou

**Affiliations:** Laboratory of Chemistry, Analysis and Design of Food Processes, Department of Food Science and Technology, School of Food Sciences, University of West Attica, 12243 Athens, Greece

**Keywords:** white mushrooms, shelf-life extension, process optimization, response surface methodology, desirability functions

## Abstract

Button mushrooms (*Agaricus bisporus*), one of the most common edible mushroom species, are sensitive to damages because of the absence of a protective skin layer and have a limited shelf life. Osmotic dehydration (OD), mainly used as a pre-processing step of conventional preservation methods, has been proposed as an efficient, mild treatment to preserve mushroom superior quality. In this study, response surface methodology, coupled with a Box–Behnken design, was used to investigate the effect of glycerol concentration (30–50%), temperature (30–50 °C), and duration of osmosis (0–180 min) in order to optimize the process prior to a subsequent freezing step. For each response, including mass transfer and selected quality indices, a second-order polynomial model was developed, and all process factors were found to have a significant impact. Based on the desirability approach and pre-set criteria, optimum operating conditions were estimated, namely osmosis at 50 °C, for 120 min, with a 42% glycerol solution, and the corresponding validation experiments were performed. Based on the error estimated between experimental and predicted values, polynomial equations were found to adequately predict parameter values. Based on a shelf-life test under frozen storage, OD-treated samples retained better quality attributes compared to their untreated counterparts.

## 1. Introduction

The button mushroom (*Agaricus bisporus*) is the most common edible mushroom, which accounts for 30% of the total consumption [1]. Mushrooms are considered a nutritionally superior food, as they are considered as good source of vitamin B complex, ergo sterols and minerals such as selenium. Additionally, they provide a variety of therapeutic compounds such as triterpenoids, glycoproteins, natural antibiotics, enzymes, and enzyme inhibitors that are beneficial for human health. Additionally, they are low in fat and considered as cholesterol-free foods, offering at the same time protein of high biological value [2]. In the last decades, there has been a tendency to reduce the additives in meat products, considering health and sustainability aspects. Mushrooms are frequently used as a good replacement because of their bioactive compounds and their flavor. In this context, more and more food companies are focused on the incorporation of mushrooms in meat products in order to produce healthier muscle food products [3,4]. The most challenging part of the production of meat analogues is the texture of the final products. Most researchers support that the incorporation of mushrooms in muscle foods do not affect the texture of the final product because of their “fibrous meat-like” texture. Thus, mushrooms can be a good replacement (up to a certain percentage) in muscle foods without causing significant negative effects on their textural characteristics [5,6,7]. Nevertheless, there are publications that report a softer texture in the final products [8,9], while others have observed an improvement in texture parameters [10].

Nonetheless, mushrooms are extremely perishable, being prone to damages because of the absence of a protective skin layer. As a consequence, the shelf life of button mushrooms is limited, reaching 1–3 days under ambient storage and 5–8 days in the refrigerator [11]. The rapid degradation is mainly caused as a result of their high water content and biological procedures such as high respiration and transpiration rates, enzyme activity, neutral pH (6.0–6.5), and microbial spoilage; furthermore, mushroom flesh is affected by the level of ripeness, and it can be seriously harmed due to the absence of cuticle, which could protect it from a potential microbial attack or physical water loss [12]. Mushroom’s short shelf life is a disadvantage that limits its economic and commercial value [1,13,14]. Among the prevailing paths of degradation of white mushrooms, enzymatic browning has the greatest impact on their quality because of the melanin production, the main outcome of enzymatic browning of fruit and vegetables. The enzyme that is responsible for browning is polyphenol oxidase (PPO), which reacts with oxygen and polyphenols. In order to control browning, it is necessary to inhibit the enzyme or to eliminate one of the two substrates in the reaction (oxygen or polyphenols) [15]. As it will be described in a later section, in our study, PPO inhibition was mainly obtained by immersing mushroom pieces into a solution of citric acid (pH reduction) prior to the subsequent osmotic dehydration step (a_w_ decrease).

Osmotic dehydration (OD) is a process that consists of removing water by dipping foods in hypertonic solutions. This process reduces nutritional and sensorial losses (flavor, color, and texture) due to the low temperatures usually applied [16]. Additionally, these mild conditions make OD an energy profitable process, with energy use being two or three times less than the corresponding requirements of conventional drying [10]. The common dehydrating agents used are sucrose (in fruits) [16,17,18] and sodium chloride (in vegetables) [19,20,21]; however, the use of multi component osmotic solutions is recently demonstrated to offer important advantages [22]. Such alternative agents include erythritol, maltitol, and xylitol because of their reduced risk of dental caries and low caloric, glycemic, and insulinemic indices [23]; glycerol and sorbitol due to their ability to induce a higher level of dehydration; sorbitol and mannose for their potential prebiotic advantages [24]; maltodextrin because it has been shown to improve mass transfer and water activity decrease; and trehalose, which has a protective role during drying [25].

Apart from the type of osmotic agent, other factors that affect OD include the concentration of osmotic solution, temperature and process time, the size/geometry of food samples, agitation level of the solution, and potential application of pretreatments. Therefore, optimization of the process is essential [26]. In general, OD is being used as a pretreatment technique or as an intermediate step prior to conventional drying or freezing for the preservation of fruits and vegetables; this is necessary as it has been shown that OD as a sole treatment does not produce final products of such a low moisture content to be considered as shelf-stable [27].

OD is widely applied in the preservation of plant-origin materials due to the reduced water activity obtained [28]. There are numerous studies on its application in apples [29], pineapples [16], mango [30], bananas [31], strawberries [17], berries [32], kiwis [33], carrots [34], potatoes [35], tomatoes [36], pumpkins [37], etc. In recent years, this method has also been applied on mushrooms such as button mushrooms [38,39,40,41,42,43], oyster mushrooms [44,45], and shiitake [46]. In those published studies on mushroom species, researchers focused mostly on the effect of the osmotic agent used and the process parameters, namely the temperature and the duration of osmosis, on mass transfer characteristics and quality attributes of different mushroom species. Gonzalez-Perez et al., 2019 [47], investigated the shrinkage phenomenon and mass transfer parameters of white mushroom pilei during osmotic dehydration in brine solutions, an osmotic agent that was also used in [41,42,43,44]. In the former study [44], oyster mushrooms were osmotically treated before being dried by two alternative methods, i.e., sun drying and cabinet drying. The final, OD-dried products were assessed based on several parameters, such as the moisture content, browning index, and drying time, in order to estimate the most efficient OD-drying process. Similarly, in [45], the effect of alternative osmotic pretreatments using salt before a final drying step (applying sun, solar, and oven drying) was evaluated based on their nutritional quality. Pei et al., 2019 [40], and Xiao et al., 2020 [43], studied the OD of button mushroom slices using ultrasound-assisted osmotic dehydration either investigating the effect of different osmotic agent (sucrose, glucose and sodium chloride) on the mass transfer parameters, average density, and microstructure [40] or modeling soluble solid content based on an hyperspectral image system [43]. However, the majority of the studies have used conventional osmotic agents (most frequently sodium chloride, followed by common sugar) in various concentrations and performed a kinetic study of mass transfer during the procedure, without aiming at optimizing the process, based on some desirable attributes of the end product. Especially in the case of mushroom, studies that have used alternative osmotic agents for the OD and results on process optimization are limited. Authors in [38] provided results on process optimization (OD of white mushroom in sugar beet molasses), measuring microbiological counts, chemical composition, and mineral content as process responses.

Based on the relative literature critical review and taking into account the increasing nutritional and functional importance of mushroom species, it would be of practical use to design a frozen end product of well-retained quality and extended shelf life based on an optimized OD pretreatment. Bearing this goal in mind, in this work, appropriate experimental design schemes (Box–Behnken design) and optimizing statistical tools (response surface methodology) were applied.

Response surface methodology (RSM) is a statistical tool extensively applied for optimization studies of preservation procedures investigating the effect of the main process parameters. The major advantage of RSM is related to the large amount of information provided from a relatively small number of experiments, allowing for the estimation of both the effect of the independent variables on the response as well as the possible interactions observed [48]. Specific experimental designs have been used by some researchers to study the parameters affecting osmotic dehydration [49].

The objective of this work was to analyze the effect of OD parameters on mass transfer phenomena and important quality attributes of button mushrooms using a multi-component OD solution. Another important goal was to optimize the process by response surface methodology (RSM), applying a multi-criterion approach, to design a product of intermediate moisture, which would be further submitted to conventional freezing. The overall scope is to design and produce a frozen end product with acceptable quality and extended shelf life.

## 2. Materials and Methods

### 2.1. Sample Preparation

Fresh edible button mushrooms were purchased from local market with an initial humidity of 91.93 ± 0.4% (wet basis). The dry mass was calculated at the end of vacuum drying (after 6 h) at 70 °C (Heraeus Instruments Vacutherm, ThermoScientific, Waltham, MA, USA), according to the official method AOAC 934.06. Each measurement was performed in three replicates to calculate the average values and the corresponding standard deviations. Mushrooms of uniform size were thoroughly washed under tap water at ambient temperature to remove surface impurities, wiped gently with a blotting paper, and cut into 8 ± 0.5 mm thick slices. Mushroom samples were then immersed into a solution of citric acid of 0.2 g/100 mL for 5 min to inhibit enzymatic browning [50], drained, and immersed in freshly prepared osmotic solutions.

### 2.2. Osmotic Dehydration

The osmotic solution was prepared by mixing different proportions of glycerol and sodium chloride with tap water. Food-grade citric acid (EMSURE^®^ for Analysis ACS, ISO, Reag. Ph Eur) and food-grade glycerol (Honeywell Specialty Chemicals Seelze GmbH, Seelze, Germany) were purchased from local providers.

According to the results of preliminary experiments and previously published data [42,51,52,53], the OD conditions used included a solution temperature (30–50 °C), immersion time (20–180 min), salt concentration (NaCl 5%), and glycerol concentration (30–50%). The aim was to determine the effect of those process parameters on water loss (WL), solid gain (SG), a_w_, color, texture, salt intake, and moisture content decrease of edible button mushrooms. For each experiment, the ratio of sample to osmotic solution was maintained constant at 1:15 (*w/w*) to avoid undesirable dilution of the osmotic solution, which would lead to a local decrease of the osmotic driving force during the OD treatment.

### 2.3. Calculation for Mass Transport Parameters for Osmotic Dehydration

Mass transfer phenomena were described in terms of water loss (WL) and solid gain (SG), according to Equations (1) and (2):(1)$$\mathrm{WL}=\frac{(M_{0}-m_{0})-\left(M-m\right)}{m_{0}}$$
(2)$$\mathrm{SG}=\frac{\left(M-m_{0}\right)}{m_{0}}$$
where M_0_ is the initial mass of fresh mushroom before the osmotic process, M is the mass of mushroom pieces after time t of the osmotic process, m is the dry mass of mushrooms after time t of the osmotic process, and m_0_ is the dry mass of fresh mushroom [54].

### 2.4. Physicochemical Measurements during Osmotic Treatment

Water activity of mushroom pieces was measured by an a_w_-meter (AquaLab Dew Point Water Activity Meter 4TE, METERGroup, Inc., Pullman, WA, USA), and the soluble solids’ content (expressed by °Brix) of the osmotic solution was determined by a hand-held refractometer (Atago, Master refractometer, Yorii, Japan). The color of mushroom samples was instrumentally determined using a tristimulus chromatometer (model CR-400, Minolta, Tokyo, Japan), calibrated with a white standard plate (L*: 97.83, a*: −0.45, b*: +1.88). The CIELAB color scale was applied based on coordinates (L*, a*, b*) being directly read from the instrument. The total color change ΔE and hue angle were calculated according to Equations (3) and (4):(3)$$\Delta E=\sqrt{\left(L_{t}^{*}-L_{0}^{*}\right)-\left(a_{t}^{*}-a_{0}^{*}\right)-\left(b_{t}^{*}-b_{0}^{*}\right)}$$
(4)$$h^{*}=\tan^{-1}\left(\frac{b^{*}}{a^{*}}\right)$$
where ΔE is the total color change; L*, a*, and b* are the luminosity, redness, and yellowness of the samples, respectively; and h* represents the hue angle. Subscripts “t” and “0” refer to time t and zero time, respectively [55]. All measurements were performed in triplicate.

A Texture Analyzer (TA-XT2i of Stable Micro Systems, Godalming, England) was used for texture analysis of all the samples, and a TPA (Texture Profile Analysis) test was carried out. The test was performed on a non-lubricated flat platform using a 6 mm cylindrical compression probe and a 25 kg load cell, under the following instrument parameters: pre-test speed—5 mm/s; test speed—2 mm/s; and post-test speed—5 mm/s at 50% deformation. Texture characteristics such as firmness, elasticity, cohesiveness, and chewiness were calculated [56]. All measurements were performed in triplicate.

### 2.5. Experimental Design

Box–Behnken design of three factors and three levels including fifteen experiments formed by three central points was implemented. The experimental data were fitted to a second-order polynomial model in order to describe the response variables Y (WL, SG, a_w_, MC (moisture content), %NaCl, ΔΕ, L/L_0_) in relation with the factor variables X_i_ (Χ_1_: temperature, X_2_: glycerol concentration, and X_3_: osmotic dehydration time).
(5)$$Y=\alpha_{0}+\sum \alpha_{i}X_{i}+\sum \alpha_{\mathrm{ii}}X_{i}^{2}+\sum \alpha_{\mathrm{ij}}X_{i}X_{j}$$
where α_0_ is the constant, and α_i_ represents the linear, α_ii_ the quadratic, and α_ij_ the interaction effects of the factors. Similar polynomial equations were also obtained for all other measured indices, namely WL, SG, MC, NaCl, ΔΕ, L/L_0_, and a_w_ change.

Optimization of the process was based on the implementation of appropriate desirability functions, as proposed by [57,58]. According to this approach, each ith response is described by a function, d_i_, where the value of d_i_ ranges from 0 to 1. In this study, the main aim of the optimization of osmotic dehydration process was to produce a final mushroom piece of intermediate moisture and with a well-preserved color, possibly destined to a further freezing preservation step. The goal of RSM application, coupled with proper desirability functions, was to find the levels of process variables, namely osmotic solution concentration, osmotic temperature, and osmotic dehydration time, which would give minimum a_w_, maximum lightness preservation (L/L_0_), and a minimum ∆E (total color difference). Therefore, the desirability function for L/L_0_ is defined as follows:(6)$$d_{1,i}=\{\begin{array}{c} 0\;\;\;\;\;\;\;\;\;\;\;\;\;\;\;\;\;\;\;y_{i}<W\\ \left(\frac{y_{i}-W}{U-W}\right)\;\;\;\;\;\;\;\;\;\;\;\;\;W\leq y_{i}\leq U\\ 1\;\;\;\;\;\;\;\;\;\;\;\;\;\;\;\;\;\;\;y_{i}>U \end{array}$$
where U represents the target value of the ith response (here equals 1), and the term represents the lower acceptable limit for that response (here equals 0.8, based on sensory rejection of the samples). In the case of a_w_ or ∆E where the goal is to obtain minimum values, the corresponding desirability functions are defined as follows:(7)$$d_{2,i}=\{\begin{array}{c} 1\;\;\;\;\;\;\;\;\;\;\;\;\;\;\;\;\;\;\;y_{i}<W\\ \left(\frac{U-y_{i}}{U-W}\right)\;\;\;\;\;\;\;\;\;\;\;\;\;W\leq y_{i}\leq U\\ 0\;\;\;\;\;\;\;\;\;\;\;\;\;\;\;\;\;\;\;y_{i}>U \end{array}$$
where W and U are the lower and upper limits of the independent variables, respectively. In our case, when assessing ∆E, W equals 0, and U is set at 10 based on sensory rejection. When referring to a_w_, in the corresponding desirability function, W equals a_w_,_min_, and U is set at the value of 1.

After evaluating the specific functions for each ith response and for the selected criteria (minimum water activity and total color difference coupled with maximum lightness preservation), a total function is defined to describe the overall requirements for osmotic dehydration optimization:(8)$$d_{\mathrm{overall},i}={\left(d_{1,i}^{r_{1}}\cdot d_{2,i}^{r_{2}}\cdot d_{3,i}^{r_{3}}\right)}^{1/\left(r_{1}+r_{2}+r_{3}\right)}$$
where the r_1_, r_2_, and r_3_ represent the importance and the relative “weight” assigned to each response (here assigned the same weight, namely r_1_ = r_2_ = r_3_ = 1).

The optimal conditions obtained by the RSM procedure, coupled with a proper desirability approach, were verified by an independent experiment.

### 2.6. Frozen Storage—Determination of Color Change and Drip Loss

To estimate the effect of osmosis on mushrooms during subsequent frozen storage, a kinetic experiment was conducted including the monitoring of some representative quality parameters (color and drip loss). For this purpose, osmotically treated (at optimum conditions) and control (untreated) samples were stored at −18 °C for ~6 months using shield packaging (PET 12/ PE 60–450 mm). The main goal of this part of the study, which is still in progress, is to verify the superior quality retention and the improved stability of the pretreated frozen samples against their untreated counterparts.

As an indicative marker of quality of frozen products, drip loss after thawing was estimated by means of Equation (9) [59]. For this purpose, frozen mushroom pieces were placed on a pre-weighed absorbent paper, allowed to thaw at ambient temperature, and finally, drip loss was calculated by weighing the absorbent paper.
(9)$$\mathrm{DL}=\frac{w_{t}-w_{0}}{w_{s}}\times 100\%$$
where w_0_ is the weight of the dry absorbent paper (g), w_t_ is the weight of the wet absorbent paper at time t (g), and w_s_ is the weight of the frozen sample (g). Three replicate runs were carried out for each test.

### 2.7. Statistical Analysis

Polynomial equations provided by RSM methodology were statistically analyzed applying analysis of variance (ANOVA), and significant differences of averages of measured values were assessed by Tukey’s HSD test at the probability level *p* < 0.05 (STATISTICA 12.0, Stat. Soft. Inc., Tulsa, OK, USA). Additionally, multivariate analysis was conducted applying principal component analysis (PCA) on mass transfer factors to further investigate correlations between properties evaluated. The analysis of Box–Behnken design and dependent variable optimization using the desirability functions tool was applied with the Minitab^®^ (DOE-response surface application, Minitab^®^ 17.1.0, Philadelphia, PA, USA).

## 3. Results and Discussion

### 3.1. Mass Transfer during Osmotic Dehydration of Button Mushrooms

In Figure 1, a_w_ change vs. process time is presented for all experimental conditions, grouped depending on the process factor modified; in each a_i_ plot, temperature is kept fixed, allowing for glycerol concentration to change, whereas in each b_i_ plot, glycerol concentration is kept fixed, allowing for temperature to change. Osmotic dehydration significantly lowered water activity as time, temperature, and glycerol concentration increased, as shown in Figure 1. a_w_ value reached an equilibrium minimum value approximately after 100 min during OD at mild conditions (30–40% concentration and 30–40 °C). Conversely, during OD at high process temperature (50 °C) and glycerol concentration (50%), a more intense a_w_ decrease was observed, and the equilibrium, observed at lower values around 0.87, was not perceived before 120 min. This is in agreement with results presented by [39,60]. The water activity of osmotically dehydrated mushrooms (OD time of 120 min) ranged from 0.8709 to 0.9312 (depending on the glycerol concentration and temperature), indicating that OD did not lead to microbiologically stable products [61].

Moisture content (MC), water loss (WL), and solid gain (SG) were used to evaluate the mass transfer phenomena during osmotic treatment. Water loss (WL) and solid gain (SG) obtained during the osmotic dehydration of mushrooms were calculated (using Equations (1) and (2), respectively). Results on the WL and SG of the white mushrooms during OD at 30, 40, and 50 °C are depicted in Figure 2 and Figure 3, respectively. WL and SG values increased significantly up to the first 100 min of OD for all measured mushroom pieces, while for longer durations, the system seemed to equilibrate. A similar pattern in the increase of WL and SG has been also described in studies on green figs [62] and on banana slices [60].

The moisture content (MC) of osmotically dehydrated mushrooms ranged from 61.12 to 76.93% after 120 min of osmosis (Figure 4). The highest water content value was recorded at the end of dehydration at 30 °C with 30% glycerol and the lowest at 40 °C with a concentration of 50%. The factor that most affected the moisture content of mushroom samples was the glycerol concentration. Moreover, the increase in immersion time led to higher moisture loss, with most samples reaching an equilibrium value after 100 min of osmosis. This is in agreement with results presented by [38,63]. In the former study [63], authors obtained the same type of curves when mackerel samples were osmotically treated in ternary solutions, mainly containing glycerol and NaCl. One should notice that in our case, the highest glycerol concentration, which led to an increased osmotic gradient, caused a more intense decrease of the final moisture content (reaching the lowest value of approximately 61% (w.b.).

As far as salt increase is concerned, the increase in immersion time led to higher salt intake, while after the first 120 min of osmosis, the absorption continued at a reduced rate. The highest NaCl absorption was recorded to be 3.63% at the end of sample dehydration at 50 °C with 30% glycerol and the lowest, 1.89%, at the end of the OD process at 30 °C with 30% glycerol. Therefore, the factor that most affected salt intake was the temperature of osmotic dehydration. That can be easily observed in Figure 5. The measurements in diagram “a”, where the temperature is constant, and the glycerol concentration is altered, are not differentiated, while in “b”, where the glycerol concentration is constant, and the osmosis temperature is varied, the measurements are separated. Although %NaCl intake is expected to be hindered by the high viscosity of the 50% glycerol solution, this effect was not observed at temperatures above 40 °C, probably to the opposite (diluting) effect of increased temperature on the viscosity. Nonetheless, from Figure 3 and Figure 5, one could correlate glycerol uptake to salt absorption. As discussed in [64], an antagonistic effect of sugar concentration and salt during the OD of plant tissues may occur, leading to a decreased salt diffusion through the plant tissue due to a barrier formation by a carbohydrate of larger molecular weight than NaCl. This might also be the case for glycerol, as can be seen in Figure 3b and Figure 5b (e.g., case of 40% glycerol at 50 °C). Based on this assumption, NaCl, having a low molecular weight (58.4 g/mol) can penetrate into the cell, leading to the decrease of the osmotic pressure gradient, while glycerol remains mainly in the extracellular space. This change in the driving force of the phenomenon (osmotic pressure change) improves the release of the water, and an enhanced water loss rate is observed, as shown in Figure 2c, thereby improving the process rate of the mass transfer for water loss for the tissues.

#### Principal Component Analysis (PCA)

To assess potential correlations and spot similarities/differences between the analyzed mushroom samples, principal component analysis (PCA) was performed in terms of properties associated to mass transport phenomena [65]. PCA was carried out on the depended values a_w_, %MC, %NaCl, WL, SG, and °Brix obtained at different immersion times, temperatures, and %glycerol. This type of analysis aims at reducing the dimensions of the data set obtained without risking the loss of useful information and assists in assessing the potential relationship among different parameters [66].

As seen in Figure 6, in the loading plot (a), the first factor accounted for 73.02% and the second for 18.46%, giving a total of 91.48% of the explained variance. As can be observed both from Figure 6a and Table 1, the first principal component was highly correlated with WL, SG, and %NaCl, while it was negatively correlated with moisture content and water activity. The second principal component was highly and positively correlated with °Brix, while it was negatively correlated with water loss and %NaCl enrichment. From a physical viewpoint, the mass transport properties WL and %NaCl were positively correlated and were placed close to each other on the biplot (Figure 6a), indicating that these parameters showed similar type of behavior, as already discussed in the previous Section 3.1. These responses were negatively correlated with a_w_ and moisture content and positioned far away from each other on biplot. This may be attributed to the fact that, as the OD proceeds, the parameters of a_w_ and moisture content reduce, whereas WL and %NaCl tend to increase, with a similar rate.

Based on PCA score plot (Figure 6b), the OD-pretreated mushroom pieces could be divided in six major groups, numbered from 1 to 6, and shown in different colors: (1-red color) treated samples with 50% glycerol at t_0_, (2-red color) treated samples with 50% glycerol at all three temperatures, (3-black color) treated samples with 40% glycerol at t_0_, (4-black color) treated samples with 40% glycerol in all three temperatures, (5-green color) treated samples with 30% glycerol at t_0_, and (6-green color) treated samples with 30% glycerol at all three temperatures. The above grading confirmed that the most effective factor of osmosis was glycerol concentration, followed by the duration of osmosis, while temperature of the osmotic solution seems to be the least effective. These results are in an agreement with the findings of [50,52]. Moreover, in Figure 6, it can be observed that group 2 has the highest values of °Brix and SG and the lowest of a_w_ and %MC. Group 6 has the lowest values of °Brix and SG. Group 4 has intermediary values. Groups 1, 3, and 5 have the highest values of a_w_ and %MC and the lowest of SG, WL, and %NaCl.

### 3.2. Quality Evaluation during Osmotic Dehydration of Button Mushrooms

Samples that were treated with 50% glycerol and at 50 °C (the most severe conditions) exhibited the most important changes in overall color (ΔΕ), indicating that extreme conditions negatively affect the color of the samples (Figure 7).

Samples that were processed in solutions of 30% glycerol concentration and samples that were processed at 30 °C maintained their hardness to a greater extent (Figure 8).

### 3.3. Determination of Parameter Interactions during Osmotic Dehydration (OD)

Based on Box–Behnken design, a total number of 15 experiments was performed with different combinations of process variables to assess the effect of independent variables (glycerol concentration, osmotic temperature, and process time) on the responses measured (WL, SG, a_w_, MC, and %NaCl) and optimize the procedure based on pre-set criteria. Applying RSM principles, the coefficients calculated for the second-order polynomial equations (Equation (5)) are depicted in Table 2 for all parameters measured, indicating the effect of processing time, processing temperature, and glycerol concentration. The asterisk sign in Table 2 accounts for a *p*-value < 0.05 and actually indicates which coefficients (contribution of each factor: linear, quadratic, interaction) are statistically significant at a confidence level of 95%. In terms of the regression coefficients, WL values are mostly affected by temperature (a1) based on the higher values of the corresponding factors (Table 3) and much less by glycerol concentration (a2) and time duration (a3). Temperature and OD solution concentration have a significant effect on a_w_ and loss of lightness (L/L_0_). In all other cases, one of the two mentioned factors had a significant effect, with process time playing a weak role (in the time interval 60–120 min, selected for RSM optimization procedure). Regarding synergistic effects, only interactions between glycerol concentration with temperature were found to have a significant effect on SG. Glycerol concentration has a negative effect on WL, whereas temperature seems to also have a negative impact on water content and water activity.

In order to assess how well the model represents the experimental data, a statistical analysis with the use of ANOVA revealed that polynomial models obtained can be generally considered as appropriate based on the R^2^(adj.) values calculated (Table 2). Additionally, the low values of the probability factor (*p* < 0.05) obtained show that the fitted models are considered statistically significant, representing the data for all response factors studied [67].

### 3.4. Optimization and Validation of Process Conditions Based on Mass Transfer and Quality Requirements

In order to find the OD process parameter values that meet the pre-defined criteria (based on factors a_w_, ∆E, and L/L_0_), the desirability function method was implemented, and the corresponding profiles of composite desirability are depicted in Figure 9. For the implementation of the methodology, the levels for each of the operational conditions (temperature, OD time, % glycerol concentration) were allowed to assume values within the range applied during the experimental procedure. The optimum operating conditions for process temperature, immersion time, and concentration were 50 °C, 120 min, and 42%, respectively. In Table 3, theoretical values of the dependent variables are estimated at those optimum conditions, calculated out of the second-order polynomial equations developed. The regression models and the theoretical calculations presented in Table 3 were validated by performing an independent experiment at the optimum predicted conditions (repeated three times). The predictions were experimentally verified through independent experiments, performed in triplicate, with a deviation not exceeding in most cases a ±18% compared to the predicted values of factors studied. In Figure 9 (left part), the desirability plot of the specific process, with the criteria set for the optimization, is illustrated. Individual (signaled with the letter “d” for each response) and composite desirability (signaled with the letter “D” for the integrated response) depict how well a combination of variables meets the goals defined for the responses. Individual desirability (d) assesses to what extent the settings optimize a single response (‘y” is the value of the response in question), whereas the composite parameter (“D”) assesses how well the settings optimize an integrated criterion. According to the algorithm followed in this work, composite desirability is calculated as the weighted geometric mean of the individual desirabilities for the selected responses. In our case, the composite desirability (close to 0.6) indicates that the settings seem to meet to a satisfactory degree the combination of criteria required. Nonetheless, the individual desirability (“d”) indicates that the process parameter values estimated are more effective at minimizing color change and water activity than at retaining the initial mushroom lightness. If one should want to emphasize the retention of the initial white color (lightness) over the other criteria, the settings should be properly re-adjusted, assigning a different weight to the target factor so as to recalculate different optimized parameters, which would provide different values for the individual and the composite desirability. In Figure 9 (right part), indicative photos of OD mushroom pieces are shown to have a more realistic illustration of their appearance at the optimized conditions.

### 3.5. Assessment of Storage Stability under Frozen Conditions

Osmotically dehydrated (OD) samples preserve their lightness during storage at −18 °C much better than the control samples (Figure 10). Moreover, OD samples show lower values in ΔΕ, while control samples have consistently high values from the eighth day of storage, showing a significant modification of their initial color.

Drip loss ranged from 27.20% at day 8 to 46.74% at day 167 for the control samples and from 11.32% to 20.56% for their osmotically dehydrated (OD) counterparts (Figure 11). Control samples have almost double percentage drip loss compared to OD samples for almost all days of storage. Consequently, osmotic dehydration retains to a much greater extent the sensory characteristics of white mushrooms, preventing samples from shrinkage after thawing and maintaining their color.

## 4. Conclusions

Based on the results of the current study, it was found that the osmotic pretreatment can significantly affect mass transfer and quality characteristics of mushroom samples before the following step of a traditional freezing process. The RSM methodology was effective in optimizing process parameters for osmotic dehydration of white mushroom in an osmotic solution of glycerol and NaCl with concentrations of 30, 40, and 50% glycerol; at a solution temperature of 30, 40, and 50 °C; and immersion time up to 180 min. Besides designing and producing intermediate moisture foods that would be better preserved under frozen storage, it is worthy also investigating whether this mild pretreatment could also assist in reducing the necessary freezing time and the required energy for water removal of the subsequent conventional freezing process. The application of Box–Behnken design as a basis of RSM, combined with the appropriate desirability functions, led to the estimation of optimum conditions of the OD process. Second-order equations were found to describe well the effect of the OD processing factors investigated (i.e., temperature and duration of OD process, OD solution concentration) on most of mass transfer parameters and quality attributes. Based on the results of frozen storage obtained, this study could serve as a basis, and extended testing is necessary in order to quantitatively assess the extension of shelf life obtained for frozen mushrooms that are osmotically pretreated.

## Figures and Tables

**Figure 1 foods-11-02354-f001:**
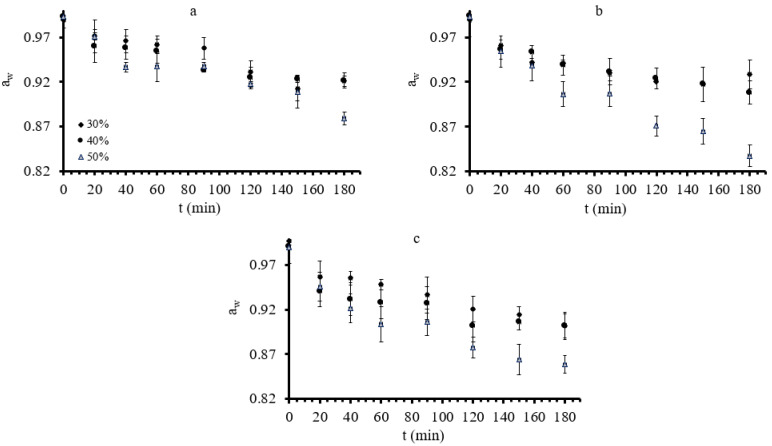
(**a**–**c**): a_w_ during osmotic dehydration at 30, 40, and 50 °C, respectively, at all three different glycerol concentrations. Points represent average values, and error bars represent the standard deviation, as calculated from the three measurements.

**Figure 2 foods-11-02354-f002:**
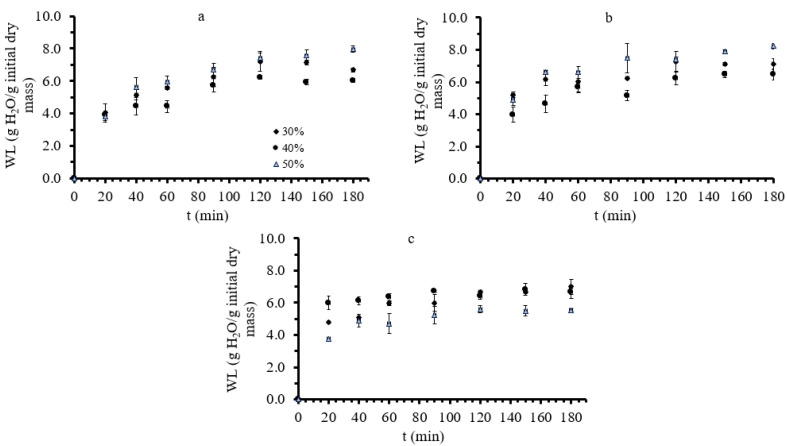
(**a**–**c**): Water loss (WL, g water/g initial dry mass) during osmotic dehydration at 30, 40, and 50 °C, respectively, at all three different glycerol concentrations. Points represent average values, and error bars represent the standard deviation, as calculated from the three measurements.

**Figure 3 foods-11-02354-f003:**
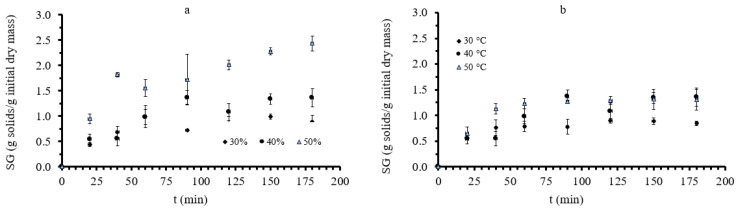
(**a**): Solid gain (SG, g solids/g initial dry mass) during osmotic dehydration at 40 °C, at all three different glycerol concentrations and (**b**) solid gain during osmotic dehydration at 40% glycerol concentration, at all three different temperatures. Points represent average values, and error bars represent the standard deviation, as calculated from the three measurements.

**Figure 4 foods-11-02354-f004:**
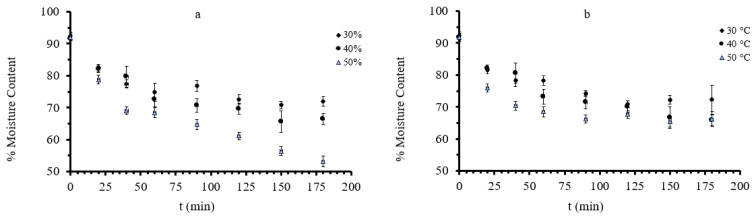
(**a**): %MC (moisture content on a wet basis) during osmotic dehydration at 40 °C, at all three different glycerol concentration and (**b**) %MC during osmotic dehydration at 40% glycerol concentration, at all three different temperatures. Points represent average values, and error bars represent the standard deviation, as calculated from the three measurements.

**Figure 5 foods-11-02354-f005:**
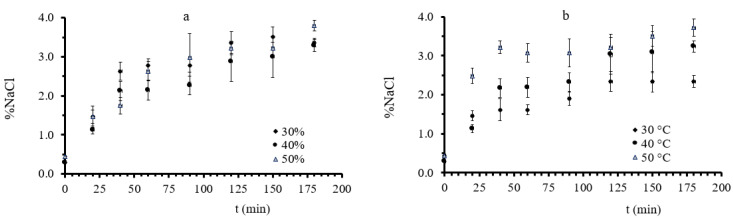
(**a**): %NaCl during osmotic dehydration at 40 °C, at all three different glycerol concentration and (**b**) %NaCl during osmotic dehydration at 40% glycerol concentration, at all three different temperatures. Points represent average values, and error bars represent the standard deviation, as calculated from the three measurements.

**Figure 6 foods-11-02354-f006:**
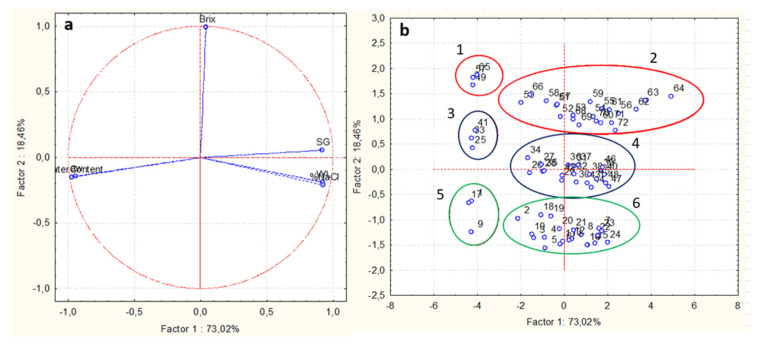
Principal component analysis (PCA) to investigate the interrelationship between water loss (WL), solid gain (SG), moisture content (MC), °Brix, %NaCl, and water activity (a_w_) with the osmotic solution formulations (osmotic concentration) and process conditions (temperature, time). (**a**): Loading plot. (**b**): Score plot, where the distinct sample groups are numbered (1 to 6) and shown in different colors.

**Figure 7 foods-11-02354-f007:**
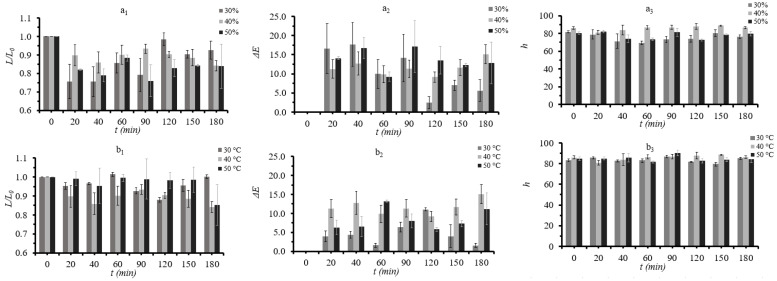
(**a_1_**): Relative lightness change (L/L_0_), (**a_2_**): color difference (ΔΕ), and (**a_3_**): hue angle (h) during osmotic dehydration at 40 °C, at all three different glycerol concentration and (**b_1_**): lightness (L), (**b_2_**): color difference (ΔΕ), and (**b_3_**): hue angle (h) during osmotic dehydration at 40% glycerol concentration, at all three different temperatures. Points represent average values, and error bars represent the standard deviation from the three measurements.

**Figure 8 foods-11-02354-f008:**
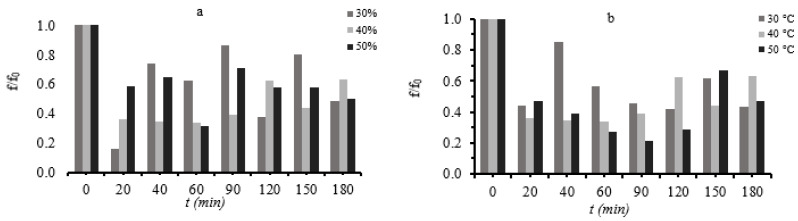
(**a**): Relative hardness change (f/f_0_) during osmotic dehydration at 40 °C, at all three different glycerol concentration and (**b**): hardness (f) during osmotic dehydration at 40% glycerol concentration, at all three different temperatures.

**Figure 9 foods-11-02354-f009:**
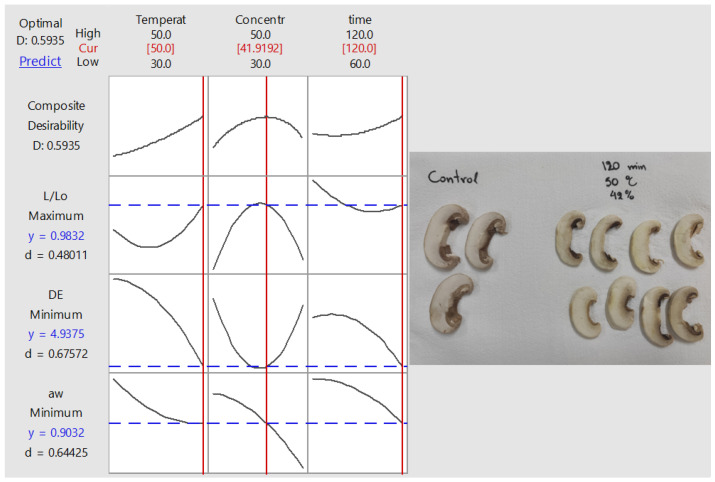
Desirability plot of variables and illustration of mushroom samples at the optimized conditions.

**Figure 10 foods-11-02354-f010:**
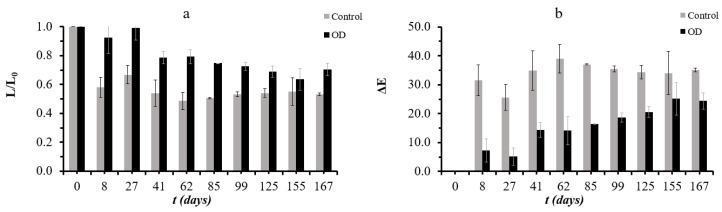
(**a**): Lightness maintenance (L/L_0_) and (**b**): color difference (ΔΕ) during storage under frozen conditions. In (**a**,**b**), points represent average values, and error bars represent the standard deviation from the three measurements.

**Figure 11 foods-11-02354-f011:**
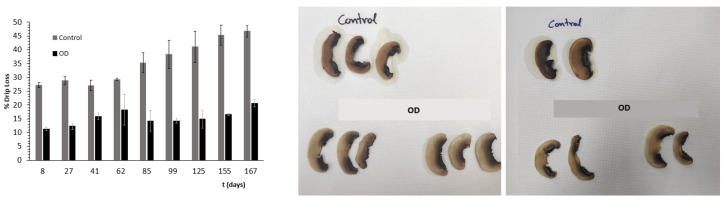
%Drip loss during storage under frozen conditions. Points represent average values, and error bars represent the standard deviation from the three measurements, and photos show the appearance of samples after 60 days (on the **left**) and 83 days (on the **right**) of frozen storage.

**Table 1 foods-11-02354-t001:** Factor coordinates of the variables, based on correlations (all). Numbers in bold depict a strong correlation.

	Factor 1	Factor 2	Factor 3	Factor 4	Factor 5	Factor 6
%MC	**−0.9734**	−0.1497	0.0700	0.0595	−0.0792	0.1238
a_w_	**−0.9381**	−0.1480	0.2405	0.1547	−0.0993	−0.0804
WL	**0.9211**	−0.1891	0.1162	0.2811	0.1500	0.0267
SG	**0.9169**	0.0533	0.3468	−0.1753	−0.0704	0.0214
%NaCl	**0.9241**	−0.2127	−0.1429	0.1086	−0.2618	0.0005
°Brix	0.0377	**0.9907**	0.0193	0.1185	−0.0506	0.0107

**Table 2 foods-11-02354-t002:** Statistical analysis of the factors of polynomial equations (RSM) with variance analysis (ANOVA).

Coefficient	% MC	a_w_	°Brix	WL	SG	L/Lo	ΔΕ	%NaCl
Constant a_0_	117.8113 *	0.915714 *	−21.0144 *	0.338394 *	−7.47012 *	0.246284 *	27.69284 *	−5.08150
a_1_	−1.4216 *	−0.007706 *	0.9035 *	0.474699	0.29925 *	0.004986 *	0.64629	0.35387 *
a_2_	−0.1856	0.007221 *	1.3535 *	−0.272870 *	0.11342	0.039845 *	−2.69240 *	0.02108
a_3_	0.0578	0.000843	−0.0591	0.023802	−0.00463	−0.005140	0.55270	−0.02118
a_11_	0.0148	0.000069	−0.0075	−0.003324	−0.00257 *	−0.000127	−0.00507	−0.0038 *
a_22_	0.0029	−0.000085 *	−0.0042	0.005273	−0.00039	−0.000441 *	0.02897	0.00051
a_33_	−0.0004	−0.000007	−0.0001	0.000182	−0.00003	0.000033	−0.00182	0.00013
a_12_	−0.0047	−0.000049	−0.0100 *	−0.003453	−0.00218 *	0.000025	0.00736	−0.00170
a_13_	0.0019	−0.00000	0.0011	−0.000902	0.00007	0.000025	−0.00620	0.00028
a_23_	−0.0033	−0.000001	0.0006	−0.000122	0.00023	−0.000050	0.00052	−0.00061
R^2^	0.869	0.843	0.983	0.752	0.801	0.721	0.698	0.795

X_1_, temperature (°C); X_2_, % glycerol concentration; X_3_, duration of osmosis (min); * *p*-value < 0.05; values assigned an asterisk are statistically significant coefficients at a level of 95%.

**Table 3 foods-11-02354-t003:** Predicted and experimental values for the responses at optimum conditions.

	Predicted Value	Experimental Value	% Error
a_w_	0.8981	0.9173 ± 0.01	2.09
WL	5.7817	5.6622 ± 0.15	−2.11
SG	1.4393	1.2091 ± 0.19	−19.04
DE	5.0417	6.0783 ± 2.42	16.94
L/L_0_	0.9898	0.9202 ± 0.03	−7.57
NaCl	3.1319	2.9220 ± 0.15	−7.18
°Brix	35.4648	36.5 ± 0.87	2.84
% MC	66.7719	68.5142 ± 1.54	2.54
f/f_0_	0.3792	0.4569 ± 0.15	17.01

## Data Availability

Data is contained within the article.

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
