# Peer review of "Optimization of Osmotic Dehydration of White Mushrooms by Response Surface Methodology for Shelf-Life Extension and Quality Improvement of Frozen End-Products"

_foods, 2022, doi:10.3390/foods11152354_

Round 1
Reviewer 1 Report
Agaricus bisporus has high relevance for protein rich diets, but is has sensitivity during processing. Pre-processing steps before preservation technologies plays significant role in the quality of the final product. Osmotic dehydration is a viable and promising pre-processing method, but optimization of the process parameters are needed to achieve high quality and shelf-life of the product. Authors used response surface methodology to optimize the temperature, time of the dehydration stage and glycerol concentration, respectively. RSM can be considered as a powerful tool for optimization of process parameters of OD. Therefore, the topic of the manuscript can be considered as interesting for the readers. The manuscript is generally well structured. Introduction summarizes well the relevance of the study. Applied methods are adequate to sample characteristics and the main aims of the research. Materials and methods are described clearly, and in details. The manuscript contains interesting results that are valuable not just for the science but also for the practice. Results are discussed in details with relevant references.
Comments, suggestions:
Abstract should summarize the main essence of the study and the main results/establishments. Therefore, I suggest giving the optimum parameters in the Abstract, as well.
Authors mentioned results of previous work focused on the optimization of process parameters (line 88-94). Please highlight the novelties of the present study in the Introduction section.
The visibility of Figure 1 and 2 is poor. Please improve it.
Please provide standard deviation for data presented in Figure 11.
Please check the typos in the manuscript (see in line 349: ‘3.4Ο. ptimization..’, for instance).
Author Response
Agaricus bisporus has high relevance for protein rich diets, but is has sensitivity during processing. Pre-processing steps before preservation technologies plays significant role in the quality of the final product. Osmotic dehydration is a viable and promising pre-processing method, but optimization of the process parameters are needed to achieve high quality and shelf-life of the product. Authors used response surface methodology to optimize the temperature, time of the dehydration stage and glycerol concentration, respectively. RSM can be considered as a powerful tool for optimization of process parameters of OD. Therefore, the topic of the manuscript can be considered as interesting for the readers. The manuscript is generally well structured. Introduction summarizes well the relevance of the study. Applied methods are adequate to sample characteristics and the main aims of the research. Materials and methods are described clearly, and in details. The manuscript contains interesting results that are valuable not just for the science but also for the practice. Results are discussed in details with relevant references.
Reply: We would like to thank the Reviewer for his/her comments and valuable suggestions that will certainly assist us in improving our submitted manuscript. We incorporated all corrections in the initial manuscript and we replied to both Reviewers’ comments, one by one. Text added in the initial manuscript is highlighted with YELLOW and lines in our replies refer to lines of the REVISED manuscript. Minor modifications throughout the text have been performed to meet the similarity percentage required and numbering of references in the list, at the end of the manuscript, has been changed, due to the incorporation of new references. All figures have been appropriately corrected, in terms of their resolution, and they have been uploaded in a .zip file (with an improved analysis >2000 dpi).
Comments, suggestions:
Abstract should summarize the main essence of the study and the main results/establishments. Therefore, I suggest giving the optimum parameters in the Abstract, as well.
Reply: Abstract was reconsidered, and the following sentence was added, to provide the results on optimum process conditions: “Based on the Desirability approach and particular pre-set criteria, optimum operating conditions were estimated, namely osmosis at 50°C, for 120 min, with a 42% glycerol solution, and the corresponding validation experiments were performed.” (lines 19-20)
Authors mentioned results of previous work focused on the optimization of process parameters (line 88-94). Please highlight the novelties of the present study in the Introduction section.
Reply: We thank the reviewer for this comment. Detailed explanations are provided in the Introduction part regarding the main focus of other works in this scientific field, and the novelty/originality of the present study is better elucidated. (lines 91-121).
The visibility of Figure 1 and 2 is poor. Please improve it.
Reply: Figures 1 and 2 have been appropriately corrected, in terms of their resolution. Additionally, all figures with an improved analysis (>2000 dpi) have been uploaded in a .zip file.
Please provide standard deviation for data presented in Figure 11.
Reply: Thank you for your comment, Figure 11 has been appropriately corrected, so as error bars (showing standard deviation of measurements) are included.
Please check the typos in the manuscript (see in line 349: ‘3.4Ο. ptimization..’, for instance). Reply: Text has been carefully checked and typos have been corrected.

Reviewer 2 Report
This manuscript deals with OD of button mushrooms. Although many articles have been published in the area, the authors provided almost no discussion supporting the need for another work in the same area. Structure of the manuscript is somehow strange, with parts from Materials and Methods presented in the Introduction, repetition of data in diverse Figures, bad quality of Figures, and specially Results and Discussion needs a huge improvement to be accepted for publication (Tables that are not necessary, Figures that are not presented and very very very few paragraphs of discussion for each Figure). The most appealing part of this manuscript, it was (in my opinion) on the frozen storage of OD mushrooms, unfortunately only preliminary data was presented.
Line 37: what about 'texture' as a quality parameter for meat replacement?
Line 46: 'any harm to the pile'? what does this mean?
Lines 53-55: re-write, English not clear.
After Line 82: Please add a good paragraph discussing what has been done in the area of OD of mushrooms (many articles as per your list of references) and mark the difference with the proposed work so as to support the need for a new paper on the area (originality).
Lines 89-93: please delete, paragraph seems incomplete by discussing only article (38).
Lines 99-108: Please combine to Materials and Methods, or delete. They have no place in the Introduction.
Line 123: tap water at ambient temperature?
Line 125: In my opinion is not 'producers' but 'providers', please CHANGE.
Lines 126-131: If so many papers have been published in the area, what is the difference with the present work and what is the novelty?
Lines 152, 159: if you produce the measurements in triplicate why Figures 1, 4, and 5 do not present standard deviation? Where are the statistical results and significance?
Table 1: should be deleted, no need to have this Table for the information presented in it.
IMPORTANT: Figures 1, 2, 3 are placed before being mentioned in the text, and Figures 7, 8, 9 and 11 are not even mentioned in the text!!! (???)
Figure 1 and 2: same data represented twice... Figures 1 or 2 'a' are at different temperatures as a function of concentrations, and 1 or 2 'b' are at different concentrations as a function of different temperatures... Please eliminate 'a' or 'b'
Figure 1: police for the title of the ordinate axis is different in 'b1' and 'b3' compared to the rest.
Figures 2 and 3: Please add units to the ordinate axis.
Line 253: please add ', respectively.' after Figure 3.
Line 260 and 316: Why 'seems' ??? you are not sure?
Figure 4 and 5: Please add more thorough discussion about these results.
Lines 286-294: Please add physical meaning to the obtained results. More discussion.
Figure 9: Please explain in more detail, and add discussion.
Author Response
This manuscript deals with OD of button mushrooms. Although many articles have been published in the area, the authors provided almost no discussion supporting the need for another work in the same area. Structure of the manuscript is somehow strange, with parts from Materials and Methods presented in the Introduction, repetition of data in diverse Figures, bad quality of Figures, and specially Results and Discussion needs a huge improvement to be accepted for publication (Tables that are not necessary, Figures that are not presented and very very very few paragraphs of discussion for each Figure).
Reply: We would like to thank the Reviewer for his/her comments and valuable suggestions that will certainly assist us in improving our submitted manuscript. We incorporated all corrections in the initial manuscript and we replied to both Reviewers’ comments, one by one. Text added in the initial manuscript is highlighted with YELLOW and lines in our replies refer to lines of the REVISED manuscript. Minor modifications throughout the text have been performed to meet the similarity percentage required and numbering of references in the list, at the end of the manuscript, has been changed, due to the incorporation of new references. All figures have been appropriately corrected, in terms of their resolution, and they have been uploaded in a .zip file (with an improved analysis >2000 dpi).
The most appealing part of this manuscript, it was (in my opinion) on the frozen storage of OD mushrooms, unfortunately only preliminary data was presented.
Reply: Regarding results on frozen storage, we removed the word ‘preliminary’ (special thanks to the reviewer for this point), as it caused a misunderstanding. As our goal is to test the behaviour of the OD samples as frozen end-products (as also stressed out in our manuscript title), those experiments under frozen storage are still in progress with a systematic kinetic study being conducted at several subfreezing temperatures, so as to be able to obtain a reliable shelf life model. This is better explained within the revised manuscript.
Line 37: what about 'texture' as a quality parameter for meat replacement?
Reply: We would like to thank the Reviewer for his/her useful observation. Based on the relevant literature, we agree with the reviewer that the most challenging part of the production of meat analogues, is to meet the requirements for the texture of the final product. Based on the references cited, it seems that the incorporation of mushrooms in muscle foods does not affect the texture of the final product, because of their “fibrous meat-like” texture. Thus, mushrooms can be a good replacement (up to a certain percentage) into muscle foods without causing significant negative effects on their textural characteristics. Nevertheless, some researchers reported softer texture in the final products, while others presented an improved texture. A relevant discussion has been added in the manuscript, in the Introduction section (lines 40-47) and (5) references were added [5-10].
Line 46: 'any harm to the pile'? what does this mean?
Reply: We rephrased the sentence, so as to provide clearly the context (lines 54-55) and one reference was added.
Lines 53-55: re-write, English not clear.
Reply: We rephrased the sentence, so as to provide clearly the context (lines 62-65).
After Line 82: Please add a good paragraph discussing what has been done in the area of OD of mushrooms (many articles as per your list of references) and mark the difference with the proposed work so as to support the need for a new paper on the area (originality).
Reply: We thank the reviewer for this comment. Detailed explanations are provided in the Introduction part regarding the main focus of other works on mushrooms in this scientific field, and the novelty/originality of the present study is better elucidated. (lines 91-121).
Lines 89-93: please delete, paragraph seems incomplete by discussing only article (38).
Reply: We agree, the paragraph has been removed from the revised manuscript.
Lines 99-108: Please combine to Materials and Methods, or delete. They have no place in the Introduction.
Reply: Text has been re-organized, following the suggestions of the reviewer.
Line 123: tap water at ambient temperature?
Reply: This information has been added within the revised manuscript.
Line 125: In my opinion is not 'producers' but 'providers', please CHANGE.
Reply: We agree, text has been corrected.
Lines 126-131: If so many papers have been published in the area, what is the difference with the present work and what is the novelty?
Reply: Detailed explanations are provided in the Introduction part regarding the main focus of other works on mushrooms in this scientific field, and the novelty/originality of the present study is better elucidated (lines 91-121). The references cited at this point mostly refer to the application of osmotic dehydration on fruits and vegetables. Mushroom studies are fewer and those studying process optimization even fewer. As discussed in the Introduction part, our goal is to proceed a step further by studying the effect of OD on the stability of frozen mushroom samples.
Lines 152, 159: if you produce the measurements in triplicate why Figures 1, 4, and 5 do not present standard deviation? Where are the statistical results and significance?
Reply: Thank you for your comment, Figures 1, 4 and 5 have been appropriately corrected, so as error bars (showing standard deviation of measurements) are included.
Table 1: should be deleted, no need to have this Table for the information presented in it. Reply: Table 1 has been removed, the enumeration of Tables has been accordingly modified.
IMPORTANT: Figures 1, 2, 3 are placed before being mentioned in the text, and Figures 7, 8, 9 and 11 are not even mentioned in the text!!! (???)
Reply: We apologize for this mistake, text has been accordingly corrected.
Figure 1 and 2: same data represented twice... Figures 1 or 2 'a' are at different temperatures as a function of concentrations, and 1 or 2 'b' are at different concentrations as a function of different temperatures... Please eliminate 'a' or 'b'
Reply: Taking into account that the effect of glycerol concentration is more important, we present only those figures (former series ‘a’), and we eliminated those of series ‘b’.
Figure 1: police for the title of the ordinate axis is different in 'b1' and 'b3' compared to the rest.
Reply: We apologize for this mistake. However, taking into account your previous suggestion, we present only figures of series ‘a’, and we eliminated those of series ‘b’.
Figures 2 and 3: Please add units to the ordinate axis.
Reply: Units have been added.
Line 253: please add ', respectively.' after Figure 3.
Reply: Text has been corrected.
Line 260 and 316: Why 'seems' ??? you are not sure?
Reply: Text has been corrected.
Figure 4 and 5: Please add more thorough discussion about these results.
Reply: We thank the reviewer for this comment. This part has been enriched and extended discussion, supported by appropriate references has been added (lines 282-290 and 306-330).
Lines 286-294: Please add physical meaning to the obtained results. More discussion.
Reply: We thank the reviewer for this comment. This part has been enriched and extended discussion, supported by appropriate references have been added (lines 339-343 and 345-356).
Figure 9: Please explain in more detail, and add discussion.
Reply: We thank the reviewer for this comment. This part has been enriched and extended discussion, supported by appropriate references have been added (lines 438-456).

Round 2
Reviewer 2 Report
The article has been improved enough.